# Etiology and pattern of maxillofacial trauma

Tahir Ullah Khan[1]*, Saima Rahat[1], Zafar Ali Khan[2], Laima Shahid[1], Syeda Sabahat Banouri[1], Nawshad Muhammad[3]*

**1** Oral and Maxillofacial Surgery, Lady Reading Hospital Medical Teaching Institute, Peshawar, Pakistan, **2** Department of Oral and Maxillofacial Surgery and Diagnostic Sciences, College of Dentistry, Jouf University, Sakaka, Saudi Arabia, **3** Department of Dental Materials, Institute of Basic Medical Sciences, Khyber Medical University, Peshawar, Pakistan

* dr.tahir786@hotmail.com (TUK); nawshad.ibms@kmu.edu.pk (NM)

**Data Availability Statement:** All relevant data are within the paper.

**Funding:** The author(s) received no specific funding for this work.

## Abstract

### Introduction

Maxillofacial trauma can be limited to superficial lacerations, abrasions, and facial bone fractures. The objective of this study was to determine the etiology, pattern, and predictors of soft tissue and bony injuries.

### Materials and methods

This study was conducted in the department of maxillofacial surgery Lady Reading hospital Pakistan from Jan 2019 to June 2021. The nonprobability consecutive sampling technique was used for the selection of patients. All patients were assessed clinically and radiologically. The neurosensory examination was done for any altered sensation, anesthesia, or paresthesia. Motor nerve function was also assessed clinically. Data were analyzed using SPSS version 26. The etiology and pattern of maxillofacial trauma were stratified among age and genders using the chi-square test to see effect modifiers. Tests for regression analysis were also applied. P≤0.05 was considered significant.

### Results

A total of 253 patients meeting inclusion criteria were included in this study. The majority of these patients were males, 223 (88.1%), while only 30 (11.9%) were females. The mean age for the group was 25.4 ± 12.6 years. RTAs were the most common causes of trauma (63.6%) followed by assault (15.0%), falls (11.5%), FAIs (5.9%), and sports (0.4%). The most vulnerable skeletal part was the mandible (22.9%) followed by Zygoma (7.1%), significantly predicted by RTAs. Soft tissue laceration analysis showed a high frequency of multiple lacerations (38%) significantly predicted by FAIs. The frequency of trigeminal nerve injury was 5.5% (14 patients) and that of the facial nerve was 1.6% (4 patients). The strongest association of nerve injury was with firearm injury (47%), followed by road traffic accidents and sports injuries.

### Conclusion

Road traffic accident was the most common etiological factor and mandible fracture was commonly predicted by RTA. Trigeminal nerve injuries were common, frequency of nerve injuries was highly associated with mandible fracture and was predicted by FAI.

Competing interests: The authors have declared that no competing interests exist.

## Introduction

Trauma is the most common cause of maxillofacial injuries [1, 2]. Injuries to skeletal components, dentition as well as soft tissues of the face happen as a result of trauma to the maxillofacial region [3]. Maxillofacial injuries are increasing in frequency and severity and this can be contributed to heavy reliance on-road transportation and the increasing socio-economic activities of the population [3]. The etiology of maxillofacial trauma has changed continuously over the past three decades, and they continue to do so [4–6]. It varies by socioeconomic status, and cultural characteristics, from one geographical location to another and among different age groups [7].

Maxillofacial trauma has a multi-factorial etiology, such as road traffic accidents (RTAs), accidental falls, assaults, industrial mishaps, sports injuries, and firearm injuries (FAIs) [8–10]. The severity and pattern of the maxillofacial trauma depend on the anatomic site of trauma, the magnitude of the force, and the direction of the force delivered to the face [9, 11]. In the past, the pattern of maxillofacial trauma was very simple [6]. Oftentimes, based on etiology and mechanism of injury, facial trauma can be limited to superficial lacerations, abrasions, and facial bone fractures, and may occur in association with other systemic injuries like head, cervical spine, chest, abdomen, and extremities, thereby requiring multidisciplinary approach for their management [12, 13].

Most of the studies conducted in the local population are focused on bony fractures. There is a scarcity of data on the pattern and etiology of maxillofacial trauma focusing on bony fractures, soft tissue injuries, and especially nerve injuries which are often ignored. The objective of this study was to determine the frequency of various etiological factors, the pattern of maxillofacial trauma, and factors predicting soft tissue and hard tissue injuries.

## Materials and methods

The sample size was calculated to be 253 by using World Health Organization (WHO) sample size calculator V.2 (1.1) by taking 1.5% frequency of lacerations on the lateral orbital region from the previous study and 1.5% margins of error and 95% confidence interval [14]. Non-probability consecutive sampling method was used to select the patients. All the patients (including both genders) between the ages of 6–60 years presenting within 2 weeks of trauma to the oral and maxillofacial unit were included in this study. Patients with severe systemic injuries, previously treated for maxillofacial injuries, and having neurological diseases were excluded from the study.

After ethical approval from the research and ethical committee of the hospital [Ref: No.1-A/LRH/MTI, dated; 28-09-2018], this descriptive study was conducted in the Department of Oral and Maxillofacial Surgery, Lady Reading Hospital Peshawar from Jan 2019 to June 2021. All patients presenting within the study time meeting the inclusion criteria were included in the study. Informed written consent was taken from each patient. Details of age, gender, location of soft tissue injury, hard tissue injury, nerve injury, and etiological factors were documented on Performa. Etiological factors were divided into road traffic accidents, falls, assault, firearms, sports, and industrial mishaps.

After initial emergency management, all patients were subjected to detailed history followed by relevant extra-oral and intraoral clinical examination. Radiographic confirmation was done by orthopantomogram, paranasal sinus view, occipitomental view, submentovertex view, and computed tomographic scan, where indicated. The neurosensory examination was done by asking about any altered sensation, anesthesia, or paresthesia in the distribution of a trigeminal nerve. Touch sensation was checked by asking the patient to close his/her eyes and a piece of 3/0 Prolene was touched on the patient's face. Pain sensations were checked with help of sterile neurosensory tips. Any loss of sensation was considered a nerve injury. Motor nerve injury

examination was done by asking the patient to produce wrinkles on the forehead, closing of eyes, filling the air in the oral cavity, smile, and chin depression. Loss of ability to perform this action was considered a motor nerve injury.

The collected data was entered and analyzed by using SPSS version 26 (SPSS Inc., Chicago, IL, USA). According to age patients were divided into two groups i.e. below 40 years and above 40 years. Quantitative variables like age, are presented in the form of mean and standard deviation (S.D). Qualitative variables like gender, etiology, and pattern are presented in the form of frequency and percentages. The etiology and pattern of maxillofacial trauma were stratified among age and genders using the chi-square test to see effect modifiers. Regression analysis was done to determine the factors that predict bony, soft tissue, and nerve injuries. P≤0.05 was considered significant.

## Results

The majority of these 253 cases of facial trauma were males, 223 (88.1%), while only 30 (11.9%) were females (Table 1). The mean age for the group was 25.4 ± 12.6 years, ranging from 6 to 60 years. The age difference between the genders was not statistically significant ($p$ = 0.577). The cause of injury did not differ significantly between genders ($p$ = 0.179), although the association of cause with dichotomized age groups (below 40 years vs. 40 years and above) was statistically significant ($p$ = .042).

### Analyses by gender

Gender had a significant association with only one (fall) of the six causes (road traffic accident, fall, assault, firearm, sports, and industrial accident), two-sample tests of proportion $z$ = 2.19, $p$ = .030. Regarding outcomes, none of the traumatic injuries (bone, soft tissue, trigeminal nerve, facial nerve) had a significant association with gender individually, with $p$ values of 0.191, 0.987, 0.158, and 0.460 respectively.

### Analyses by age

The mean age for the group was 25.4 ± 12.6 years, ranging from 6 to 60 years. The distribution of age had a significant positive skew (skewness = 0.9). The variance of age did not differ significantly by gender (p = 0.338) or outcomes but was significantly unequal between the various categories of causes. The age difference between the genders was not statistically significant (p = 0.577). Among the six causes, only two had a significant association with age: Road traffic accident victims were significantly younger (p = 0.030) while firearm, assault victims were significantly older. Among the various outcomes, only trigeminal nerve injury had a significant association with age.

### Analyses by cause

Road traffic accidents were the most common cause of trauma, accounting for 63.6% of cases, while assault (15.0%), falls (11.5%), and firearm injuries (5.9%) accounted for the majority of the remaining cases, with sports injury being the least common cause (0.4%). Cause of injury did not differ significantly between genders (Pearson Chi-Square: $\chi 2(5)$ = 7.61, p = 0.179), although the association of cause with dichotomized age groups (below 40 years vs. 40 years and above) was statistically significant (p = 0.042). The majority of nerve injuries (47%) were due to firearm injuries.

**Table 1. Univariate analysis of predictors and outcome variable.**

| Bone Trauma: | n | % | CI (95%) |
|---|---|---|---|
| No fracture | 83 | 32.8 | 27.1–39.0% |
| Maxilla | 15 | 5.9 | 3.4–9.6% |
| Zygoma | 18 | 7.1 | 4.3–11.0% |
| Mandible | 58 | 22.9 | 17.9–28.6% |
| Dentoalveolar | 23 | 9.1 | 5.8–13.3% |
| Multiple Bones | 43 | 17.0 | 12.6–22.2% |
| Frontal Bone | 1 | 0.4 | 0.0–2.2% |
| Systemic Injuries | 12 | 4.7 | 2.5–8.1% |
| Soft Tissue Injury: | | | |
| No laceration | 84 | 33.2 | 27.4–39.4% |
| Forehead | 27 | 10.7 | 7.2–15.1% |
| Infraorbital Region | 12 | 4.7 | 2.5–8.1% |
| Cheek | 23 | 9.1 | 5.8–13.3% |
| Lower Lip and Chin | 30 | 11.9 | 8.1–16.5% |
| Intraoral | 33 | 13.0 | 9.2–17.8% |
| Upper Lip | 6 | 2.4 | 0.9–5.1% |
| Multiple Lacerations | 38 | 15.0 | 10.8–20.0% |
| **Nerve Injury**: | | | |
| Trigeminal Nerve | 14 | 5.5 | 3.0–9.1% |
| Facial Nerve | 4 | 1.6 | 0.4–4.0% |
| **Age**: | | | |
| 6–20 Years | 21 | 8.3 | 5.2–12.4% |
| 21–35 Years | 94 | 37.2 | 31.2–43.4% |
| 36–50 Years | 93 | 36.8 | 30.8–43.0% |
| $\geq$ 51 Years | 45 | 17.8 | 13.3–23.1% |
| **Gender**: | | | |
| Male | 223 | 88.1 | 83.5–91.8% |
| Female | 30 | 11.9 | 8.1–16.5% |
| **Cause**: | | | |
| Road Traffic Accident | 161 | 63.6 | 57.4–69.6% |
| Fall | 29 | 11.5 | 7.8–16.0% |
| Assault | 38 | 15.0 | 10.8–20.0% |
| Firearm | 15 | 5.9 | 3.4–9.6% |
| Sports | 1 | 0.4 | 0.0–2.2% |
| Industrial | 9 | 3.6 | 1.6–6.6% |

## Analysis by outcome

Patients with facial bone injuries were significantly younger than those with other bone injuries. The association of age with bone injuries, using the ANOVA procedure, was marginally significant, while no such association existed between age and soft tissue injuries (p = 0.502).

**Bivariate analyses.** (t-tests, tests of proportions, ANOVA procedures, Chi-Square tests; Table 2).

Bivariate analysis was done for gender, age, cause, and outcome, and details are presented in Table 2.

**Table 2. Bivariate analyses.**

| Outcomes | Age (T-Test, p values) | Gender (Test of proportions, p values) | RTA | Fall | Assault | Firearm | Sports-Indus. | | | | | | | |
|---|---|---|---|---|---|---|---|---|---|---|---|---|---|---|
| **Bone Injury:** | | | | | | | | | | | | | | |
| Maxilla | ANOVA F(7, 245) = 2.56, p = .014 | .612 | chi-square(7) = 10.4, p = .002 | .855 | chi-square(7) = 22.7, p = .166 | .014 | chi-square(7) = 8.6 p = .280 | .151 | Chi square(7) = 33., p < .001 | .093 | chi-square(7) = 9. p = .223 | .901 | chi-square(7) = 4.3, p = .743 | .418 |
| Zygoma | | .137 | | .106 | | .072 | | .414 | | .244 | | .269 | | .717 |
| Mandible | | .592 | | .326 | | .553 | | .869 | | .256 | | .024 | | .587 |
| Dentoalveolar | | .120 | | .623 | | .772 | | .105 | | .781 | | .207 | | .919 |
| Multiple Bones | | .243 | | .034 | | .021 | | .124 | | .003 | | .304 | | .264 |
| Frontal Bone | | * | | * | | * | | * | | * | | * | | * |
| Other Systemic | | .002 | | .149 | | .823 | | .727 | | .506 | | .718 | | .425 |
| **Soft Tissue Injury:** | | | | | | | | | | | | | | |
| Forehead | ANOVA F(7, 245) = 0.91, p = .344 | .120 | chi-square(7) = 5.4, p = .617 | .899 | chi-square(7) = 13.8, p = .054 | .442 | chi-square(7) = 2.6, p = .921 | .484 | chi-square(7) = 14.1, p = .049 | .590 | chi-square(7) = 18.0, p = .012 | .168 | chi-square(7) = 6.1, p = .532 | .944 |
| Infraorbital | | .331 | | .699 | | .402 | | .202 | | .321 | | .373 | | .472 |
| Cheek | | .653 | | .065 | | .010 | | .803 | | .344 | | .130 | | .019 |
| Lower Lip & Chin | | .983 | | .142 | | .398 | | .732 | | .416 | | .855 | | .853 |
| Intraoral | | .846 | | .960 | | .020 | | .899 | | .010 | | .450 | | .771 |
| Upper Lip | | .914 | | .712 | | .310 | | .686 | | .297 | | .534 | | .615 |
| Multiple Lacerations | | .290 | | .788 | | .947 | | .844 | | .070 | | < .001 | | .650 |
| **Trigeminal Injury** | | | | | | | | | | | | | | |
| Infraorbital | ANOVA F(4, 248) = 3.74, p = .006 | .044 | chi-square(4) = 2.0, p = .089 | .460 | chi-square(4) = 18.0, p = .001 | .008 | chi-square(4) = 2.03, p = .730 | .392 | chi-square(4) = 2.1, p = .714 | .573 | chi-square(4) = 58.5, p = < .001 | .104 | chi-square(4) = 22.6, p = < .001 | .029 |
| Inferior Alveolar | | .020 | | .603 | | .687 | | .609 | | .551 | | .008 | | .773 |
| Mental | | .860 | | .408 | | .003 | | .416 | | .342 | | < .001 | | < .001 |
| Multiple Branches | | .031 | | .523 | | .272 | | .531 | | .464 | | < .001 | | .724 |
| **Facial Nerve Injury** | | | | | | | | | | | | | | |
| Temporal | .141 | .603 | .060 | .086 | .551 | .008 | .773 | | | | | | | |
| Multiple Branches | * | * | * | * | * | * | * | | | | | | | |
| All Branches | * | * | * | * | * | * | * | | | | | | | |

*Only one case

## Multivariate analysis

On Multinomial Logistic Regression, compared with the risk of non-firearm assault as baseline, the risk of trauma from firearm injury significantly increased with age (Relative Risk Ratio = 1.05, test for Regression Coefficient being zero, $z = 2.32$, $p = .020$) while gender did not predict any specific kind of trauma (RTA: $p = .732$, Fall: $p = .112$, Firearm: $p = .979$, Sports & Industrial: $p = .986$).

## Multivariate analysis (Multinomial Logistic Regression, Table 3)

Compared with the risk of non-firearm assault as baseline, age substantially predicted Zygomatic bone injury (Relative Risk Ratio (RRR) = 1.1, test for Regression Coefficient being zero $z = 2.73$, $p = .006$), multiple facial bones injury (RRR = 1.04, $z = 2.22$, $p = .027$), and general skeletal injuries (RRR = 1.1, $z = 3.23$, $p = .001$). Age also predicted forehead soft tissue injuries significantly (RRR = 1.04, $z = 2.21$, $p = .027$). Gender did not predict any of the outcomes with statistical significance. Among various causes, road traffic accidents significantly predicted zygomatic (RRR = 16.3, $z = 2.53$, $p = .011$) as well as mandibular injuries (RRR = 4.1, $z = 2.74$, $p = .006$), while falls did not predict any of the outcomes with statistical significance. Firearm

**Table 3. Multivariate analyses (Multinomial Logistic Regression).**

| Outcomes | Predictors | | | | | | |
|---|---|---|---|---|---|---|---|
| | Age | Gender | RTA | Fall | Assault | Firearm | Sports-Indus. |
| | p values for tests (coefficient = 0), after adjusting for all other variables | | | | | | |
| Bone Injury: Base outcome: no injury | | | | | | | |
| Maxilla | .109 | .793 | .989 | 1.000 | Base level | .989 | .999 |
| Zygoma | .006 | .989 | .011 | .384 | " | .994 | .064 |
| Mandible | .563 | .874 | .006 | .165 | " | .004 | .034 |
| Dentoalveolar | .993 | .332 | .092 | .110 | " | .993 | .167 |
| Multiple Bones | .027 | .056 | .982 | .983 | " | .981 | .981 |
| Frontal Bone | * | * | * | * | * | * | * |
| Other Systemic | .001 | .519 | .055 | .688 | Base level | .203 | .054 |
| Soft Tissue Injury: Base outcome: no lacerations | | | | | | | |
| Forehead | .027 | .850 | .501 | .761 | " | .993 | .979 |
| Infraorbital | .637 | .778 | .948 | .990 | " | .995 | .995 |
| Cheek | .554 | .986 | .463 | .858 | " | .186 | .258 |
| Lower Lip & Chin | .598 | .260 | .965 | .985 | " | .349 | .981 |
| Intraoral: | .204 | .956 | .981 | .982 | " | .982 | .982 |
| Upper Lip: | .580 | .842 | .992 | .992 | " | 1.000 | 1.000 |
| Multiple Lacerations: | .306 | .644 | .067 | .218 | " | .003 | .444 |
| Trigeminal Injury: Base outcome: no nerve injury | | | | | | | |
| Infraorbital: | .193 | .998 | .996 | .642 | " | .510 | .376 |
| Inferior Alveolar: | .094 | .998 | .998 | 1.000 | " | .998 | 1.000 |
| Mental: | .184 | 1.000 | 1.000 | 1.000 | " | .997 | .997 |
| Multiple Branches: | .252 | .998 | .998 | 1.000 | " | .998 | 1.000 |
| Facial Nerve Injury: Base outcome: no nerve injury | | | | | | | |
| Temporal: | .338 | .998 | 1.000 | .998 | " | .998 | 1.000 |
| Multiple Branches: | * | * | * | * | * | * | * |
| All Branches: | * | * | * | * | * | * | * |

* Only one case

assault significantly predicted mandibular injuries among facial bones (RRR = 14.9, $z$ = 2.91, $p$ = .004) and multiple lacerations among facial soft tissue injuries (RRR = 26.4, $z$ = 2.95, $p$ = .003). The only outcome predicted by sports and industrial accidents was mandibular fractures (RRR = 13.9, $z$ = 2.12, $p$ = .034).

The most vulnerable skeletal part was the mandible, significantly predicted by road traffic accidents, firearm assaults, sports, and industrial accidents. The next one was Zygoma fracture, predicted by age (significantly higher mean age for those with injury) and road traffic accidents.

## Discussion

Maxillofacial trauma has multifactorial etiology and is one of the leading causes resulting in damage to facial soft tissues and bones [15, 16]. Maxillofacial trauma has specific characteristics, treatment modalities, and outcomes [17]. Therefore understanding the pattern and characteristics of maxillofacial injuries is of prime importance in the prevention and management of such injuries.

In our study males were dominant with a male to female ratio of 9:1. The age difference between the sex was not statistically significant. Males are at higher risk due to their greater participation in the active population, mainly in non-developed countries, which increases their exposure to risk factors such as driving vehicles, sports, an active social life, and drug use, including alcohol. However, in some regions, maxillofacial trauma is high in females probably due to changes in women's social behavior. Cultural and socioeconomic features have a significant influence on gender prevalence rates of maxillofacial injuries [1, 6].

The etiology of maxillofacial fractures has changed continuously over the past three decades, and they continue to do so [8, 18, 19]. Maxillofacial trauma has a multi-factorial etiology, such as road traffic accidents (RTAs), accidental falls, assaults, industrial mishaps, sports injuries, and firearm injuries (FAIs) [10, 20]. Reasons for the high frequency of RTA in developing countries are inadequate road safety awareness, unsuitable road conditions without expansion of the motorway network, violation of speed limit, old vehicles without safety features, not wearing seatbelts and helmets, violation of highway code, and population adherence to preventive measures is also very rare in the local population [1, 3]. The reason for accidents in our setup was due to socioeconomic conditions and violation of traffic rules whereas, in developed countries, accidents are mostly due to alcohol intoxication [4]. In our study, the cause of injury did not differ significantly between genders. Gender had a significant association with only one (fall) of the six causes (road traffic accident, fall, assault, firearm, sports, and industrial accident), it was high in patients as this group falls from a roof, trees, and cliff while playing [2, 21].

The most vulnerable skeletal part was mandible, significantly predicted by road traffic accidents, firearm assaults, sports, and industrial accidents, followed by Zygoma fracture predicted by age and road traffic accidents. This is in agreement with other reports from across Asian countries but differs from studies from the western world where nasal bone and zygomatic complex fractures were a more common occurrence [3, 6]. Contrary to our findings, Arslan ED et al observed that majority of injuries are concentrated around the middle third and upper third of the face [22].

Soft tissue laceration analysis showed a high frequency of multiple lacerations (38%), significantly predicted by FAIs. Age also predicted soft tissue injuries significantly. Soft tissue injuries did not show significant association with any of the six causes (($\chi^2$(35) = 45.1, $p$ = .117), individual $p$ values for tests of proportions ranging from the smallest .092 for firearm injury to the largest .5556 for falls. HM Hussaini et al found that upper and lower lips were the most

affected area in soft tissue injury [23]. Here, soft-tissue injuries commonly involved the lower third of the face, particularly the lips and chin. Contrary to our findings, Udeabor S et al observed that soft tissue overlying Zygoma was more affected by contusions and abrasions instead of lacerations which is consistent with our observation as well [12]. Other concomitant systemic injuries were recorded to be relatively low in our study as compared to other reported studies. We observed a high incidence of concomitant injuries to upper and lower limbs. This finding is in line with previous studies that showed a high incidence of injury to the limbs [6, 10, 24].

The strongest association of nerve injury was with firearm injury (47%), followed by road traffic accidents and sports injuries. Among the nerve injuries, the only significant association of facial nerve injury was with firearm injury while trigeminal nerve injuries were significantly less common in road traffic accidents, while significantly more common in firearm injuries. The only injury in the sports category was trigeminal nerve injury. Facial nerve injuries were more common in the temporal branch. Our results on facial nerve injuries are not consistent with previous studies. B Poorian et al and Tahir et al found that the most common involved branch was marginal mandibular [24, 25].

## Limitations of the study

We did not evaluate the level of nerve injuries and the lack of follow-up was another important limitation. So, we could not conclude whether these nerve impairments were transient or permanent.

## Conclusion

A road traffic accident was the most common etiological factor and mandible fracture was common. Trigeminal nerve injuries were common and the frequency of nerve injuries was high in relation to mandible fractures. The most vulnerable skeletal part was the mandible, significantly predicted by road traffic accidents, firearm assaults, sports, and industrial accidents. The next common fracture was Zygoma, predicted by age and road traffic accidents.

## Acknowledgments

We are thankful to Prof. Zahid Nazar (Prof. of Psychiatry MTI Lady Reading Hospital) and Prof Bashart Hussain (Biostatistician) for their valuable input and guidance to prepare this manuscript.

## Author Contributions

**Conceptualization:** Laima Shahid.

**Data curation:** Tahir Ullah Khan, Saima Rahat, Laima Shahid.

**Formal analysis:** Tahir Ullah Khan, Saima Rahat, Laima Shahid, Syeda Sabahat Banouri, Nawshad Muhammad.

**Funding acquisition:** Tahir Ullah Khan, Syeda Sabahat Banouri.

**Investigation:** Zafar Ali Khan, Syeda Sabahat Banouri.

**Methodology:** Saima Rahat, Zafar Ali Khan.

**Supervision:** Tahir Ullah Khan.

**Writing – review & editing:** Nawshad Muhammad.

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
