## [Decision Letter · Decision Letter 0]

2 Aug 2022

PONE-D-22-14079ETIOLOGY AND PATTERN OF MAXILLOFACIAL TRAUMA- REGRESSION ANALYSISPLOS ONE

Dear Dr. Muhammad,

Thank you for submitting your manuscript to PLOS ONE. After careful consideration, we feel that it has merit but does not fully meet PLOS ONE’s publication criteria as it currently stands. Therefore, we invite you to submit a revised version of the manuscript that addresses the points raised during the review process.

Please consider all comments

We look forward to receiving your revised manuscript.

Kind regards,

Ahmed Mancy Mosa, Ph.D.

Academic Editor

PLOS ONE

Journal Requirements:

- https://www.sciencedirect.com/science/article/abs/pii/S2212555815000149?via%3Dihub

- https://www.scielo.br/j/jaos/a/WYNhnPJxkqj78bCTgHXjSrf/?format=pdf&lang=en

- https://pubmed.ncbi.nlm.nih.gov/23594725/

- https://ayubmed.edu.pk/JAMC/PAST/15-2/Shahid%20Maxillofacial.htm

In your revision ensure you cite all your sources (including your own works), and quote or rephrase any duplicated text outside the methods section. Further consideration is dependent on these concerns being addressed.

Reviewers' comments:

Reviewer's Responses to Questions

**Comments to the Author**

1. Is the manuscript technically sound, and do the data support the conclusions?

Reviewer #1: Partly

Reviewer #2: Yes

2. Has the statistical analysis been performed appropriately and rigorously? 

Reviewer #1: Yes

Reviewer #2: Yes

3. Have the authors made all data underlying the findings in their manuscript fully available?

Reviewer #1: No

Reviewer #2: Yes

4. Is the manuscript presented in an intelligible fashion and written in standard English?

Reviewer #1: Yes

Reviewer #2: Yes

5. Review Comments to the Author

Reviewer #1: - In your study titled 'ETIOLOGY AND PATTERN OF MAXILLOFACIAL TRAUMA-REGRESSION ANALYSIS', 'regression analysis' is a common analysis method. Therefore, I recommend using the title as follows, excluding that analysis. 'ETIOLOGY AND PATTERN OF MAXILLOFACIAL TRAUMA'

- INTRODUCTION

The content of the last paragraph of the introduction is very similar to that of Objectives. Therefore, we recommend that you delete Objectives or describe it in a different content from the previous paragraph.

- MATERIALS AND METHODS:

In your study, the study subjects were limited to patients who visited the hospital for 2 weeks,

It is thought that the patients selected for the study were followed up from January 2019 to June 2021.

However, this part is not fully understood with the research method described by the authors. Therefore, to help the reader understand, please amend the following sentence to describe the content more accurately.

'All the patients (including both genders) between age 6-60 years presenting within 2 weeks of trauma to oral and maxillofacial unitwere included in this study'

'this descriptive study was conducted in the Department of Oral and Maxillofacial Surgery, Lady Reading Hospital Peshawar from Jan 2019 to June 2021.'

Also, in the title, regression analysis was mentioned as an analysis method, but the content was omitted in the research method.

- RESULTS

Currently table 1 is less readable. Authors are advised to put effort into presenting the table in a more neatly organized way to increase the readability of the reader.

Also, in presenting univariate results, rather than explaining each content in detail by dividing it into gender, age, and cause, detailed results are presented using tables, and in the case of content explained in writing, more impactful content is presented. I think it would be preferable.

- DISCUSSION

Is it correct to consider only Gender as 'Cultural and socioeconomic factors' in your study?

Although gender is a factor that includes a relatively social concept when compared with sex, it seems unreasonable to think that 'Cultural and socioeconomic factor' is considered only with gender.

Considering that it is an etiology study, we recommend that you add analysis by considering more diverse variables.

Reviewer #2: I would just like to point out a few things:

1. The Materials and Methods section on the Abstract is insufficient.There should be a sentence or two about how the data was collected and analyzed.

2. In the Introduction section I would recommend that the Objective be merged with the last paragraph and not written as a separate section.

3. There are a few grammatical errors that require correction throughout the manuscript.

6. PLOS authors have the option to publish the peer review history of their article (what does this mean?). If published, this will include your full peer review and any attached files.

Reviewer #1: No

Reviewer #2: No

---

## [Author Response · Author response to Decision Letter 0]

31 Aug 2022

Response to reviewer comments

We appreciate the comments of reviewers and modified the article concerning their suggestions 

Reviewer #1: - In your study titled 'ETIOLOGY AND PATTERN OF MAXILLOFACIAL TRAUMA-REGRESSION ANALYSIS', 'regression analysis' is a common analysis method. Therefore, I recommend using the title as follows, excluding that analysis. 'ETIOLOGY AND PATTERN OF MAXILLOFACIAL TRAUMA'

Reply: We changed the title as suggested. 

- INTRODUCTION

The content of the last paragraph of the introduction is very similar to that of Objectives. Therefore, we recommend that you delete Objectives or describe it in a different content from the previous paragraph.

Reply. It has been modified as suggested

- MATERIALS AND METHODS:

In your study, the study subjects were limited to patients who visited the hospital for 2 weeks,

It is thought that the patients selected for the study were followed up from January 2019 to June 2021.

However, this part is not fully understood with the research method described by the authors. Therefore, to help the reader understand, please amend the following sentence to describe the content more accurately.

Reply: It has been amended as suggested.

'All the patients (including both genders) between age 6-60 years presenting within 2 weeks of trauma to oral and maxillofacial unitwere included in this study'

'this descriptive study was conducted in the Department of Oral and Maxillofacial Surgery, Lady Reading Hospital Peshawar from Jan 2019 to June 2021.'

Also, in the title, regression analysis was mentioned as an analysis method, but the content was omitted in the research method.

Reply: It has been readdressed as suggested.

- RESULTS

Currently table 1 is less readable. Authors are advised to put effort into presenting the table in a more neatly organized way to increase the readability of the reader.

Also, in presenting univariate results, rather than explaining each content in detail by dividing it into gender, age, and cause, detailed results are presented using tables, and in the case of content explained in writing, more impactful content is presented. I think it would be preferable.

Reply: It has been readdressed as suggested.

- DISCUSSION

Is it correct to consider only Gender as 'Cultural and socioeconomic factors' in your study?

Although gender is a factor that includes a relatively social concept when compared with sex, it seems unreasonable to think that 'Cultural and socioeconomic factor' is considered only with gender.

Considering that it is an etiology study, we recommend that you add analysis by considering more diverse variables.

Reply: It has been readdressed as suggested.

Reviewer #2: I would just like to point out a few things:

1. The Materials and Methods section on the Abstract is insufficient.There should be a sentence or two about how the data was collected and analyzed.

Reply: It has been readdressed as suggested.

2. In the Introduction section I would recommend that the Objective be merged with the last paragraph and not written as a separate section.

Reply: It has been readdressed as suggested.

3. There are a few grammatical errors that require correction throughout the manuscript.

Reply: We have thoroughly revised the manuscript as suggested 

6. PLOS authors have the option to publish the peer review history of their article (what does this mean?). If published, this will include your full peer review and any attached files.

---

## [Decision Letter · Decision Letter 1]

19 Sep 2022

ETIOLOGY AND PATTERN OF MAXILLOFACIAL TRAUMA

PONE-D-22-14079R1

Dear Dr. Muhammad,

We’re pleased to inform you that your manuscript has been judged scientifically suitable for publication and will be formally accepted for publication once it meets all outstanding technical requirements.

Kind regards,

Ahmed Mancy Mosa, Ph.D.

Academic Editor

PLOS ONE

Additional Editor Comments (optional):

Reviewers' comments:

Reviewer's Responses to Questions

**Comments to the Author**

1. If the authors have adequately addressed your comments raised in a previous round of review and you feel that this manuscript is now acceptable for publication, you may indicate that here to bypass the “Comments to the Author” section, enter your conflict of interest statement in the “Confidential to Editor” section, and submit your "Accept" recommendation.

Reviewer #2: (No Response)

2. Is the manuscript technically sound, and do the data support the conclusions?

Reviewer #2: (No Response)

3. Has the statistical analysis been performed appropriately and rigorously? 

Reviewer #2: (No Response)

4. Have the authors made all data underlying the findings in their manuscript fully available?

Reviewer #2: (No Response)

5. Is the manuscript presented in an intelligible fashion and written in standard English?

Reviewer #2: (No Response)

6. Review Comments to the Author

Reviewer #2: (No Response)

7. PLOS authors have the option to publish the peer review history of their article (what does this mean?). If published, this will include your full peer review and any attached files.

Reviewer #2: No

---

## [Editor Report · Acceptance letter]

21 Sep 2022

PONE-D-22-14079R1 

Etiology and pattern of maxillofacial trauma 

Dear Dr. Muhammad:

I'm pleased to inform you that your manuscript has been deemed suitable for publication in PLOS ONE. Congratulations! Your manuscript is now with our production department. 

Kind regards, 

on behalf of

Dr. Ahmed Mancy Mosa 

Academic Editor

PLOS ONE